# ALFWorld: Aligning Text and Embodied Environments for Interactive Learning

**Mohit Shridhar**[†]    **Xingdi Yuan**[♡]    **Marc-Alexandre Côté**[♡]
**Yonatan Bisk**[‡]    **Adam Trischler**[♡]    **Matthew Hausknecht**[♣]
[†]University of Washington    [♡]Microsoft Research, Montréal
[‡]Carnegie Mellon University    [♣]Microsoft Research

**ALFWorld.github.io**

## Abstract

Given a simple request like *Put a washed apple in the kitchen fridge*, humans can reason in purely abstract terms by imagining action sequences and scoring their likelihood of success, prototypicality, and efficiency, all without moving a muscle. Once we see the kitchen in question, we can update our abstract plans to fit the scene. Embodied agents require the same abilities, but existing work does not yet provide the infrastructure necessary for both reasoning abstractly and executing concretely. We address this limitation by introducing ALFWorld, a simulator that enables agents to learn abstract, text-based policies in TextWorld (Côté et al., 2018) and then execute goals from the ALFRED benchmark (Shridhar et al., 2020) in a rich visual environment. ALFWorld enables the creation of a new BUTLER agent whose abstract knowledge, learned in TextWorld, corresponds directly to concrete, visually grounded actions. In turn, as we demonstrate empirically, this fosters better agent generalization than training only in the visually grounded environment. BUTLER's simple, modular design factors the problem to allow researchers to focus on models for improving every piece of the pipeline (language understanding, planning, navigation, and visual scene understanding).

## 1 Introduction

Consider helping a friend prepare dinner in an unfamiliar house: when your friend asks you to clean and slice an apple for an appetizer, how would you approach the task? Intuitively, one could reason abstractly: (1) find an apple (2) wash the apple in the sink (3) put the clean apple on the cutting board (4) find a knife (5) use the knife to slice the apple (6) put the slices in a bowl. Even in an unfamiliar setting, abstract reasoning can help accomplish the goal by leveraging semantic priors. Priors like *locations of objects* – apples are commonly found in the kitchen along with implements for cleaning and slicing, *object affordances* – a sink is useful for washing an apple unlike a refrigerator, *pre-conditions* – better to wash an apple before slicing it, rather than the converse. We hypothesize that, learning to solve tasks using abstract language, unconstrained by the particulars of the physical world, enables agents to complete embodied tasks in novel environments by leveraging the kinds of semantic priors that are exposed by abstraction and interaction.

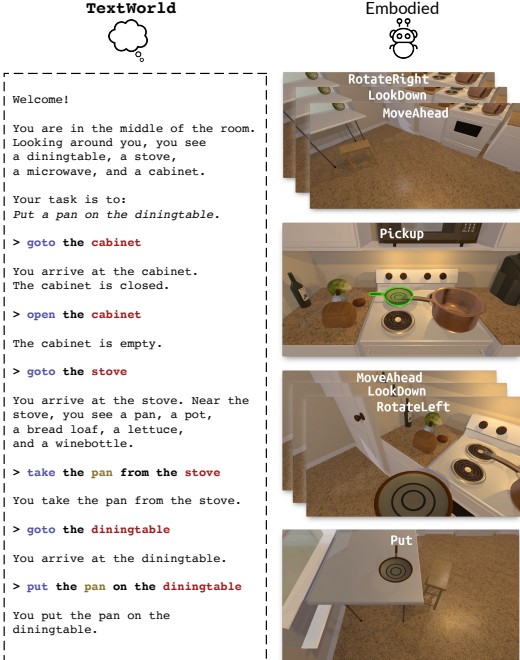

Figure 1: ALFWorld: Interactive aligned text and embodied worlds. An example with high-level text actions (left) and low-level physical actions (right).

To test this hypothesis, we have created the novel ALFWorld framework, the first interactive, parallel environment that aligns text descriptions and commands with physically embodied robotic simulation. We build ALFWorld by extending two prior works: TextWorld (Côté et al., 2018) - an engine for interactive text-based games, and ALFRED (Shridhar et al., 2020) - a large scale dataset for vision-language instruction following in embodied environments. ALFWorld provides two views of the same underlying world and two modes by which to interact with it: TextWorld, an abstract, text-based environment, generates textual observations of the world and responds to high-level text actions; ALFRED, the embodied simulator, renders the world in high-dimensional images and responds to low-level physical actions as from a robot (Figure 1).[1] Unlike prior work on instruction following (MacMahon et al., 2006; Anderson et al., 2018a), which typically uses a static corpus of cross-modal expert demonstrations, we argue that aligned parallel environments like ALFWorld offer a distinct advantage: they allow agents to *explore, interact*, and *learn* in the abstract environment of language before encountering the complexities of the embodied environment.

While fields such as robotic control use simulators like MuJoCo (Todorov et al., 2012) to provide infinite data through interaction, there has been no analogous mechanism – short of hiring a human around the clock – for providing linguistic feedback and annotations to an embodied agent. TextWorld addresses this discrepancy by providing programmatic and aligned linguistic signals during agent exploration. This facilitates the first work, to our knowledge, in which an embodied agent learns the meaning of complex multi-step policies, expressed in language, directly through interaction.

Empowered by the ALFWorld framework, we introduce BUTLER (**B**uilding **U**nderstanding in **T**extworld via **L**anguage for **E**mbodied **R**easoning), an agent that first learns to perform abstract tasks in TextWorld using Imitation Learning (IL) and then transfers the learned policies to embodied tasks in ALFRED. When operating in the embodied world, BUTLER leverages the abstract understanding gained from TextWorld to generate text-based actions; these serve as high-level subgoals that facilitate physical action generation by a low-level controller. Broadly, we find that BUTLER is capable of generalizing in a zero-shot manner from TextWorld to unseen embodied tasks and settings. Our results show that training first in the abstract text-based environment is not only 7× faster, but also yields better performance than training from scratch in the embodied world. These results lend credibility to the hypothesis that solving abstract language-based tasks can help build priors that enable agents to generalize to unfamiliar embodied environments.

Our contributions are as follows:

§ 2 **ALFWorld environment**: The first parallel interactive text-based and embodied environment.

§ 3 **BUTLER architecture**: An agent that learns high-level policies in language that transfer to low-level embodied executions, and whose modular components can be independently upgraded.

§ 4 **Generalization**: We demonstrate empirically that BUTLER, trained in the abstract text domain, generalizes better to unseen embodied settings than agents trained from corpora of demonstrations or from scratch in the embodied world.

## 2 ALIGNING ALFRED AND TEXTWORLD

**The ALFRED dataset** (Shridhar et al., 2020), set in the THOR simulator (Kolve et al., 2017), is a benchmark for learning to complete embodied household tasks using natural language instructions and egocentric visual observations. As shown in Figure 1 (right), ALFRED tasks pose challenging interaction and navigation problems to an agent in a high-fidelity simulated environment. Tasks are annotated with a goal description that describes the objective (e.g., "put a pan on the dining table"). We consider both template-based and human-annotated goals; further details on goal specification can be found in Appendix H. Agents observe the world through high-dimensional pixel images and interact using low-level action primitives:
MOVEAHEAD, ROTATELEFT/RIGHT, LOOKUP/DOWN, PICKUP, PUT, OPEN, CLOSE, and TOGGLEON/OFF.

| Task type | # train | # seen | # unseen |
|---|---|---|---|
| Pick & Place | 790 | 35 | 24 |
| Examine in Light | 308 | 13 | 18 |
| Clean & Place | 650 | 27 | 31 |
| Heat & Place | 459 | 16 | 23 |
| Cool & Place | 533 | 25 | 21 |
| Pick Two & Place | 813 | 24 | 17 |
| All | 3,553 | 140 | 134 |

Table 1: Six ALFRED task types with heldout seen and unseen evaluation sets.

---

[1]Note: Throughout this work, for clarity of exposition, we use ALFRED to refer to both tasks and the grounded simulation environment, but rendering and physics are provided by THOR (Kolve et al., 2017).

The ALFRED dataset also includes crowdsourced language instructions like "turn around and walk over to the microwave" that explain *how* to complete a goal in a step-by-step manner. We depart from the ALFRED challenge by omitting these step-by-step instructions and focusing on the more diffcult problem of using only on goal descriptions specifying *what* needs to be achieved.

Our aligned ALFWorld framework adopts six ALFRED `task-types` (Table 1) of various difficulty levels.[2] Tasks involve first *finding a particular object*, which often requires the agent to open and search receptacles like drawers or cabinets. Subsequently, all tasks other than Pick & Place require some interaction with the object such as heating (place object in microwave and start it) or cleaning (wash object in a sink). To complete the task, the object must be placed in the designated location.

Within each task category there is significant variation: the embodied environment includes 120 `rooms` (30 kitchens, 30 bedrooms, 30 bathrooms, 30 living rooms), each dynamically populated with a set of portable `objects` (e.g., apple, mug), and static `receptacles` (e.g., microwave, fridge). For each task type we construct a larger train set, as well as seen and unseen validation evaluation sets: **(1):** seen consists of known task instances {`task-type`, `object`, `receptacle`, `room`} in `rooms` seen during training, but with different instantiations of `object` locations, quantities, and visual appearances (e.g. two blue pencils on a shelf instead of three red pencils in a drawer seen in training). **(2):** unseen consists of new task instances with possibly known `object`-`receptacle` pairs, but always in unseen `rooms` with different `receptacles` and scene layouts than in training tasks.

The seen set is designed to measure in-distribution generalization, whereas the unseen set measures out-of-distribution generalization. The scenes in ALFRED are visually diverse, so even the same task instance can lead to very distinct tasks, e.g., involving differently colored apples, shaped statues, or textured cabinets. For this reason, purely vision-based agents such as the unimodal baselines in Section 5.2 often struggle to generalize to unseen environments and objects.

**The TextWorld framework** (Côté et al., 2018) procedurally generates text-based environments for training and evaluating language-based agents. In order to extend TextWorld to create text-based analogs of each ALFRED scene, we adopt a common latent structure representing the state of the simulated world. ALFWorld uses PDDL - Planning Domain Definition Language (McDermott et al., 1998) to describe each scene from ALFRED and to construct an equivalent text game using the TextWorld engine. The dynamics of each game are defined by the PDDL domain (see Appendix C for additional details). Textual observations shown in Figure 1 are generated with templates sampled from a context-sensitive grammar designed for the ALFRED environments. For interaction, TextWorld environments use the following high-level actions:

```
goto {recep}          take {obj} from {recep}    put {obj} in/on {recep}
open {recep}          close {recep}              toggle {obj}{recep}
clean {obj} with {recep}   heat {obj} with {recep}   cool {obj} with {recep}
```

where `{obj}` and `{recep}` correspond to objects and receptacles. Note that `heat`, `cool`, `clean`, and `goto` are high-level actions that correspond to several low-level embodied actions.

**ALFWorld**, in summary, is an cross-modal framework featuring a diversity of embodied tasks with analogous text-based counterparts. Since both components are fully interactive, agents may be trained in either the language or embodied world and evaluated on heldout test tasks in either modality. We believe the equivalence between objects and interactions across modalities make ALFWorld an ideal framework for studying language grounding and cross-modal learning.

## 3 Introducing BUTLER: An Embodied Multi-task Agent

We investigate learning in the abstract language modality *before* generalizing to the embodied setting. The BUTLER agent uses three components to span the language and embodied modalities: BUTLER::Brain – the abstract text agent, BUTLER::Vision – the language state estimator, and BUTLER::Body – the low-level controller. An overview of BUTLER is shown in Figure 2 and each component is described below.

---

[2]To start with, we focus on a subset of the ALFRED dataset for training and evaluation that excludes tasks involving slicing objects or using portable container (e.g., bowls).

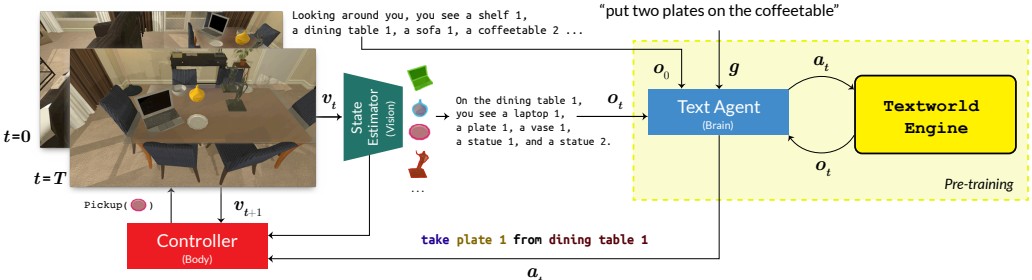

Figure 2: **BUTLER Agent** consists of three modular components. 1) BUTLER::BRAIN: a text agent pre-trained with the TextWorld engine (indicated by the dashed yellow box) which simulates an abstract textual equivalent of the embodied world. When subsequently applied to embodied tasks, it generates high-level actions that guide the controller. 2) BUTLER::VISION: a state estimator that translates, at each time step, the visual frame $v_t$ from the embodied world into a textual observation $o_t$ using a pre-trained Mask R-CNN detector. The generated observation $o_t$, the initial observation $o_0$, and the task goal $g$ are used by the text agent the to predict the next high-level action $a_t$. 3) BUTLER::BODY: a controller that translates the high-level text action $a_t$ into a sequence of one or more low-level embodied actions.

## 3.1 BUTLER::BRAIN (TEXT AGENT) : $o_0, o_t, g \rightarrow a_t$

BUTLER::BRAIN is a novel text-based game agent that generates high-level text actions in a token-by-token fashion akin to Natural Language Generation (NLG) approaches for dialogue (Sharma et al., 2017) and summarization (Gehrmann et al., 2018). An overview of the agent's architecture is shown in Figure 3. At game step $t$, the encoder takes the initial text observation $o_0$, current observation $o_t$, and the goal description $g$ as input and generates a context-

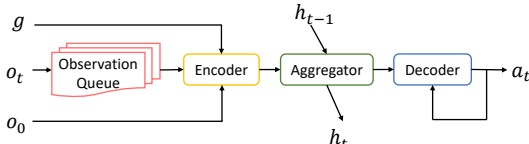

Figure 3: BUTLER::BRAIN: The text agent takes the initial/current observations $o_0/o_t$, and goal $g$ to generate a textual action $a_t$ token-by-token.

aware representation of the current observable game state. The observation $o_0$ explicitly lists all the navigable receptacles in the scene, and goal $g$ is sampled from a set of language templates (see Appendix H). Since the games are partially observable, the agent only has access to the observation describing the effects of its previous action and its present location. Therefore, we incorporate two memory mechanisms to imbue the agent with history: (1) a recurrent aggregator, adapted from Yuan et al. (2018), combines the encoded state with recurrent state $h_{t-1}$ from the previous game step; (2) an observation queue feeds in the $k$ most recent, unique textual observations. The decoder generates an action sentence $a_t$ token-by-token to interact with the game. The encoder and decoder are based on a Transformer Seq2Seq model with pointer softmax mechanism (Gulcehre et al., 2016). We leverage pre-trained BERT embeddings (Sanh et al., 2019), and tie output embeddings with input embeddings (Press and Wolf, 2016). The agent is trained in an imitation learning setting with DAgger (Ross et al., 2011) using expert demonstrations. See Appendix A for complete details.

When solving a task, an agent might get stuck at certain states due to various failures (e.g., action is grammatically incorrect, wrong object name). The observation for a failed action does not contain any useful feedback, so a fully deterministic actor tends to repeatedly produce the same incorrect action. To address this problem, during evaluation in both TextWorld and ALFRED, BUTLER::BRAIN uses Beam Search (Reddy et al., 1977) to generate alternative action sentences in the event of a failed action. But otherwise greedily picks a sequence of best words for efficiency. Note that Beam Search is not used to optimize over embodied interactions like prior work (Wang et al., 2019). but rather to simply improve the generated action sentence during failures.

## 3.2 BUTLER::VISION (STATE ESTIMATOR) : $v_t \rightarrow o_t$

At test time, agents in the embodied world must operate purely from visual input. To this end, BUTLER::VISION's language state estimator functions as a captioning module that translates visual observations $v_t$ into textual descriptions $o_t$. Specifically, we use a pre-trained Mask R-CNN detec-

tor (He et al., 2017) to identify objects in the visual frame. The detector is trained separately in a supervised setting with random frames from ALFRED training scenes (see Appendix D). For each frame $v_t$, the detector generates $N$ detections $\{(c_1, m_1), (c_2, m_2), \ldots, (c_N, m_N)\}$, where $c_n$ is the predicted object class, and $m_n$ is a pixel-wise object mask. These detections are formatted into a sentence using a template e.g., `On table 1, you see a mug 1, a tomato 1, and a tomato 2`. To handle multiple instances of objects, each object is associated with a class $c_n$ and a number ID e.g., `tomato 1`. Commands **`goto`**, **`open`**, and **`examine`** generate a list of detections, whereas all other commands generate affirmative responses if the action succeeds e.g., $a_t$: **`put mug 1 on desk 2`** $\rightarrow o_{t+1}$: `You put mug 1 on desk 2`, otherwise produce `Nothing happens` to indicate failures or no state-change. See Appendix G for a full list of templates. While this work presents preliminary results with template-based descriptions, future work could generate more descriptive observations using pre-trained image-captioning models (Johnson et al., 2016), video-action captioning frameworks (Sun et al., 2019), or scene-graph parsers (Tang et al., 2020).

### 3.3   BUTLER::BODY (CONTROLLER) : $v_t, a_t \rightarrow \{\hat{a}_1, \hat{a}_2, \ldots, \hat{a}_L\}$

The controller translates a high-level text action $a_t$ into a sequence of $L$ low-level physical actions $\{\hat{a}_1, \hat{a}_2, \ldots, \hat{a}_L\}$ that are executable in the embodied environment. The controller handles two types of commands: manipulation and navigation. For manipulation actions, we use the ALFRED API to interact with the simulator by providing an API action and a pixel-wise mask based on Mask R-CNN detections $m_n$ that was produced during state-estimation. For navigation commands, each episode is initialized with a pre-built grid-map of the scene, where each receptacle instance is associated with a receptacle class and an interaction viewpoint $(x, y, \theta, \phi)$ with $x$ and $y$ representing the 2D position, $\theta$ and $\phi$ representing the agent's yaw rotation and camera tilt. The **`goto`** command invokes an A* planner to find the shortest path between two viewpoints. The planner outputs a sequence of $L$ displacements in terms of motion primitives: MOVEAHEAD, ROTATERIGHT, ROTATELEFT, LOOKUP, and LOOKDOWN, which are executed in an open-loop fashion via the ALFRED API. We note that a given pre-built grid-map of receptacle locations is a strong prior assumption, but future work could incorporate existing models from the vision-language navigation literature (Anderson et al., 2018a; Wang et al., 2019) for map-free navigation.

## 4   EXPERIMENTS

We design experiments to answer the following questions: (1) How important is an *interactive* language environment versus a static corpus? (2) Do policies learnt in TextWorld transfer to embodied environments? (3) Can policies generalize to human-annotated goals? (4) Does pre-training in an abstract textual environment enable better generalization in the embodied world?

### 4.1   IMPORTANCE OF INTERACTIVE LANGUAGE

The first question addresses our core hypothesis that training agents in interactive TextWorld environments leads to better generalization than training agents with a static linguistic corpus. To test this hypothesis, we use DAgger (Ross et al., 2011) to train the BUTLER::BRAIN agent in TextWorld and compare it against **Seq2Seq**, an identical agent trained with Behavior Cloning from an equivalently-sized corpus of expert demonstrations. The demonstrations come from the same expert policies and we control the number of episodes to ensure a fair comparison. Table 2 presents results for agents trained in TextWorld and subsequently evaluated in embodied environments in a zero-shot manner. The agents are trained independently on individual tasks and also jointly on all six task types. For each task category, we select the agent with best evaluation performance in TextWorld (from 8 random seeds); this is done separately for each split: seen and unseen. These best-performing agents are then evaluated on the heldout seen and unseen embodied ALFRED tasks. For embodied evaluations, we also report goal-condition success rates, a metric proposed in ALFRED (Shridhar et al., 2020) to measure partial goal completion.[3]

---

[3]For instance, the task "put a hot potato on the countertop" is composed of three goal-conditions: (1) heating some object, (2) putting a potato on the countertop, (3) heating a potato and putting it on the countertop. If the agent manages to put any potato on the countertop, then $1/3 = 0.33$ goal-conditions are satisfied, and so on.

| task-type | TextWorld | | Seq2Seq | | BUTLER | | BUTLER-ORACLE | | Human Goals | |
|---|---|---|---|---|---|---|---|---|---|---|
| | seen | unseen | seen | unseen | seen | unseen | seen | unseen | seen | unseen |
| Pick & Place | 69 | 50 | 28 (28) | 17 (17) | **30 (30)** | **24 (24)** | 53 (53) | 31 (31) | 20 (20) | 10 (10) |
| Examine in Light | 69 | 39 | 5 (13) | 0 (6) | **10 (26)** | 0 (15) | 22 (41) | 12 (37) | 2 (9) | 0 (8) |
| Clean & Place | 67 | 74 | **32 (41)** | 12 (31) | 32 (46) | **22 (39)** | 44 (57) | 41 (56) | 18 (31) | 22 (39) |
| Heat & Place | 88 | 83 | 10 (29) | 12 (33) | **17 (38)** | **16 (39)** | 60 (66) | 60 (72) | 8 (29) | 5 (30) |
| Cool & Place | 76 | 91 | 2 (19) | **21 (34)** | **5 (21)** | 19 (33) | 41 (49) | 27 (44) | 7 (26) | 17 (34) |
| Pick Two & Place | 54 | 65 | 12 (23) | 0 (26) | **15 (33)** | **8 (30)** | 32 (42) | 29 (44) | 6 (16) | 0 (6) |
| All Tasks | 40 | 35 | 6 (15) | 5 (14) | **19 (31)** | **10 (20)** | 37 (46) | 26 (37) | 8 (17) | 3 (12) |

Table 2: **Zero-shot Domain Transfer**. *Left*: Success percentages of the best BUTLER::BRAIN agents evaluated purely in TextWorld. *Mid-Left*: Success percentages after zero-shot transfer to embodied environments. *Mid-Right*: Success percentages of BUTLER with an oracle state-estimator and controller, an upper-bound. *Right*: Success percentages of BUTLER with human-annotated goal descriptions, an additional source of generalization difficulty. All successes are averaged across three evaluation runs. Goal-condition success rates (Shridhar et al., 2020) are given in parentheses. The **Seq2Seq** baseline is trained in TextWorld from pre-recorded expert demonstrations using standard supervised learning. **BUTLER** is our main model using the Mask R-CNN detector and A* navigator. **BUTLER-ORACLE** uses an oracle state-estimator with ground-truth object detections and an oracle controller that directly teleports between locations.

Comparing **BUTLER** to **Seq2Seq**, we see improved performance on all types of seen tasks and five of the seven types of unseen tasks, supporting the hypothesis that interactive TextWorld training is a key component in generalizing to unseen embodied tasks. Interactive language not only allows agents to explore and build an understanding of successful action patterns, but also to recover from mistakes. Through trial-and-error the BUTLER agent learns task-guided heuristics, e.g., searching all the drawers in kitchen to look for a knife. As Table 2 shows, these heuristics are subsequently more capable of generalizing to the embodied world. More details on TextWorld training and generalization performance can be found in Section 5.1.

## 4.2 TRANSFERRING TO EMBODIED TASKS

Since TextWorld is an abstraction of the embodied world, transferring between modalities involves overcoming *domain gaps* that are present in the real world but not in TextWorld. For example, the physical size of objects and receptacles must be respected – while TextWorld will allow certain objects to be placed inside any receptacle, in the embodied world it might be impossible to put a larger object into a small receptacle (e.g. a large pot into a microwave).

Subsequently, a TextWorld-trained agent's ability to solve embodied tasks is hindered by these domain gaps. So to study the transferability of the text agent in isolation, we introduce **BUTLER-ORACLE** in Table 2, an oracle variant of BUTLER which uses perfect state-estimation, object-detection, and navigation. Despite these advantages, we nevertheless observe a notable drop in performance from **TextWorld** to **BUTLER-ORACLE**. This performance gap results from the domain gaps described above as well as misdetections from Mask R-CNN and navigation failures caused by collisions. Future work might address this issue by reducing the domain gap between the two environments, or performing additional fine-tuning in the embodied setting.

The supplementary video contains qualitative examples of the BUTLER agent solving tasks in unseen environments. It showcases 3 successes and 1 failure of a TextWorld-only agent trained on All Tasks. In "put a watch in the safe", the agent has never seen the 'watch'-'safe' combination as a goal.

## 4.3 GENERALIZING TO HUMAN-ANNOTATED GOALS

BUTLER is trained with templated language, but in realistic scenarios, goals are often posed with open-ended natural language. In Table 2, we present **Human Goals** results of BUTLER evaluated on human-annotated ALFRED goals, which contain 66 unseen verbs (e.g., 'wash', 'grab', 'chill') and 189 unseen nouns (e.g., 'rag', 'lotion', 'disc'; see Appendix H for full list). Surprisingly, we find non-trivial goal-completion rate indicating that certain categories of task, such as pick and place, are quite generalizable to human language. While these preliminary results with natural language are encouraging, we expect future work could augment the templated language with synthetic-to-real transfer methods (Marzoev et al., 2020) for better generalization.

### 4.4 To Pretrain or not to Pretrain in TextWorld?

Given the domain gap between TextWorld and the embodied world, *Why not eliminate this gap by training from scratch in the embodied world?* To answer this question, we investigate three training strategies: (i) EMBODIED-ONLY: pure embodied training, (ii) TW-ONLY: pure TextWorld training followed by zero-shot embodied transfer and

| Training Strategy | train (succ %) | seen (succ %) | unseen (succ %) | train speed (eps/s) |
|---|---|---|---|---|
| EMBODIED-ONLY | 21.6 | **33.6** | 23.1 | 0.9 |
| TW-ONLY | 23.1 | 27.1 | **34.3** | **6.1** |
| HYBRID | 11.9 | 21.4 | 23.1 | 0.7 |

Table 3: **Training Strategy Success**. Trained on All Tasks for 50K episodes and evaluated in embodied scenes using an oracle state-estimator and controller.

(iii) HYBRID training that switches between the two environments with 75% probability for TextWorld and 25% for embodied world. Table 3 presents success rates for these agents trained and evaluated on All Tasks. All evaluations were conducted with an oracle state-estimator and controller. For a fair comparison, each agent is trained for 50K episodes and the training speed is recorded for each strategy. We report peak performance for each split.

Results indicate that TW-ONLY generalizes better to unseen environments while EMBODIED-ONLY quickly overfits to seen environments (even with a perfect object detector and teleport navigator). We hypothesize that the abstract TextWorld environment allows the agent to focus on quickly learning tasks without having to deal execution-failures and expert-failures caused by physical constraints inherent to embodied environments. TextWorld training is also 7× faster[4] since it does not require running a rendering or physics engine like in the embodied setting. See Section F for more quantitative evaluations on the benefits of training in TextWorld.

## 5 Ablations

We conduct ablation studies to further investigate: (1) The generalization performance of BUTLER::BRAIN within TextWorld environments, (2) The ability of unimodal agents to learn directly through visual observations or action history, (3) The importance of various hyper-parameters and modeling choices for the performance of BUTLER::BRAIN.

### 5.1 Generalization within TextWorld

We train and evaluate BUTLER::BRAIN in abstract TextWorld environments spanning the six tasks in Table 1, as well as All Tasks. Similar to the zero-shot results presented in Section 4.1, the All Tasks setting shows the extent to which a *single* policy can learn and generalize on the large set of 3,553 different tasks, but here without having to deal with failures from embodied execution.

We first experimented with training BUTLER::BRAIN through reinforcement learning (RL) where the agent is rewarded after completing a goal. Due to the infesibility of using candidate commands or command templates as discussed in Section I, the RL agent had to generate actions token-by-token. Since the probability of randomly stumbling upon a grammatically correct and contextually valid action is very low (7.02e-44 for sequence length 10), the RL agent struggled to make any meaningful progress towards the tasks.

After concluding that current reinforcement learning approaches were not successful on our set of training tasks, we turned to DAgger (Ross et al., 2011) assisted by a rule-based expert (detailed in Appendix E). BUTLER::BRAIN is trained for 100K episodes using data collected by interacting with the set of training games.

Results in Table 4 show (i) Training success rate varies from 16-60% depending on the category of tasks, illustrating the challenge of solving hundreds to thousands of training tasks within each category. (ii) Transferring from training to heldout test games typically reduces performance, with the unseen rooms leading to the largest performance drops. Notable exceptions include heat and cool tasks where unseen performance exceeds training performance. (iii) Beam search is a key contributor to test performance; its ablation causes a performance drop of 21% on the seen split of All Tasks. (iv) Further ablating the DAgger strategy and directly training a Sequence-to-Sequence (Seq2Seq) model

---

[4]For a fair comparison, all agents in Table 3 use a batch-size of 10. THOR instances use 100MB×batch-size of GPU memory for rendering, whereas TextWorld instances are CPU-only and are thus much easier to scale up.

| | Pick & Place | | | Examine in Light | | | Clean & Place | | | Heat & Place | | | Cool & Place | | | Pick Two & Place | | | All Tasks | | |
|---|---|---|---|---|---|---|---|---|---|---|---|---|---|---|---|---|---|---|---|---|---|
| | tn | sn | un | tn | sn | un | tn | sn | un | tn | sn | un | tn | sn | un | tn | sn | un | tn | sn | un |
| BUTLER | 54 | **61** | 46 | 59 | **39** | 22 | 37 | **44** | 39 | 60 | **81** | 74 | 46 | 60 | **100** | 27 | 29 | **24** | 16 | **40** | **37** |
| BUTLER$_g$ | 54 | 43 | 33 | 59 | 31 | 17 | 37 | 30 | 26 | 60 | 69 | 70 | 46 | 50 | 76 | 27 | **38** | 12 | 16 | 19 | 22 |
| Seq2Seq | 31 | 26 | 8 | 44 | 31 | 11 | 34 | 30 | **42** | 36 | 50 | 30 | 27 | 32 | 33 | 17 | 8 | 6 | 9 | 10 | 9 |

Table 4: **Generalization within TextWorld environments:** We independently train BUTLER::BRAIN on each type of TextWorld task and evaluate on heldout scenes of the same type. Respectively, tn/sn/un indicate success rate on train/seen/unseen tasks. All sn and un scores are computed using the random seeds (from 8 in total) producing the best final training score on each task type. BUTLER is trained with DAgger and performs beam search during evaluation. Without beam search, BUTLER$_g$ decodes actions *greedily* and gets stuck repeating failed actions. Further removing DAgger and training the model in a Seq2Seq fashion leads to worse generalization. Note that tn scores for BUTLER are lower than sn and un as they were computed without beam search.

with pre-recorded expert demonstrations causes a bigger performance drop of 30% on seen split of All Tasks. These results suggest that online interaction with the environment, as facilitated by DAgger learning and beam search, is essential for recovering from mistakes and sub-optimal behavior.

## 5.2 UNIMODAL BASELINES

Table 5 presents results for unimodal baseline comparisons to BUTLER. For all baselines, the action space and controller are fixed, but the state space is substituted with different modalities. To study the agents' capability of learning a single policy that generalizes across various tasks, we train and evaluate on All Tasks. In VISION (RESNET18), the textual observation from the state-estimator is replaced with ResNet-18 `fc7` features (He et al., 2016) from the visual frame. Similarly, VISION (MCNN-FPN) uses the

| Agent | seen (succ %) | unseen (succ %) |
|---|---|---|
| BUTLER | **18.8** | **10.1** |
| VISION (RESNET18) | 10.0 | 6.0 |
| VISION (MCNN-FPN) | 11.4 | 4.5 |
| ACTION-ONLY | 0.0 | 0.0 |

Table 5: **Unimodal Baselines**. Trained on All Tasks with 50K episodes and evaluated in the embodied environment.

pre-trained Mask R-CNN from the state-estimator to extract FPN layer features for the whole image. ACTION-ONLY acts without any visual or textual feedback. We report peak performance for each split.

The visual models tend to overfit to seen environments and generalize poorly to unfamiliar environments. Operating in text-space allows better transfer of policies without needing to learn state representations that are robust to visually diverse environments. The zero-performing ACTION-ONLY baseline indicates that memorizing action sequences is an infeasible strategy for agents.

## 5.3 MODEL ABLATIONS

Figure 4 illustrates more factors that affect the performance of BUTLER::BRAIN. The three rows of plots show training curves, evaluation curves in seen and unseen settings, respectively. All experiments were trained and evaluated on All Tasks with 8 random seeds.

In the first column, we show the effect of using different observation queue lengths $k$ as described in Section 3.1, in which size 0 refers to not providing any observation information to the agent. In the second column, we examine the effect of explicitly keeping the initial observation $o_0$, which lists all the receptacles in the

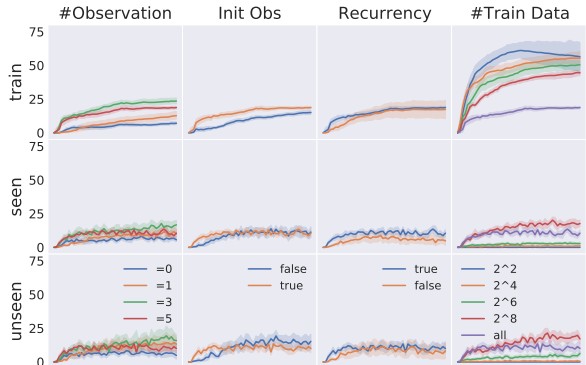

Figure 4: Model ablations on All Tasks. x-axis: 0 to 50k episodes; y-axis: normalized success from 0 to 75%.

scene. Keeping the initial observation $o_0$ facilitates the decoder to generate receptacle words more accurately for unseen tasks, but may be unnecessary in seen environments. The third column suggests that the recurrent component in our aggregator is helpful in making history-based decisions

particularly in seen environments where keeping track of object locations is useful. Finally, in the fourth column, we see that using more training games can lead to better generalizability in both seen and unseen settings. Fewer training games achieve high training scores by quickly overfitting, which lead to zero evaluation scores.

## 6 RELATED WORK

The longstanding goal of grounding language learning in embodied settings (Bisk et al., 2020) has lead to substantial work on interactive environments. ALFWorld extends that work with fully-interactive aligned environments that parallel textual interactions with photo-realistic renderings and physical interactions.

**Interactive Text-Only Environments**: We build on the work of text-based environments like TextWorld (Côté et al., 2018) and Jericho (Hausknecht et al., 2020). While these environment allow for textual interactions, they are not grounded in visual or physical modalities.

**Vision and language**: While substantial work exists on vision-language representation learning e.g., MAttNet (Yu et al., 2018b), CMN (Hu et al., 2017), VQA (Antol et al., 2015), CLEVR (Johnson et al., 2017), ViLBERT (Lu et al., 2019), they lack embodied or sequential decision making.

**Embodied Language Learning**: To address language learning in embodied domains, a number of interactive environments have been proposed: BabyAI (Chevalier-Boisvert et al., 2019), Room2Room (Anderson et al., 2018b), ALFRED (Shridhar et al., 2020), InteractiveQA (Gordon et al., 2018), EmbodiedQA (Das et al., 2018), and NetHack (Küttler et al., 2020). These environments use language to communicate instructions, goals, or queries to the agent, but not as a fully-interactive textual modality.

**Language for State and Action Representation**: Others have used language for more than just goal-specification. Schwartz et al. (2019) use language as an intermediate state to learn policies in VizDoom. Similarly, Narasimhan et al. (2018) and Zhong et al. (2020) use language as an intermediate representation to transfer policies across different environments. Hu et al. (2019) use a natural language instructor to command a low-level executor, and Jiang et al. (2019) use language as an abstraction for hierarchical RL. However these works do not feature an interactive text environment for pre-training the agent in an abstract textual space. Zhu et al. (2017) use high-level commands similar to ALFWorld to solve tasks in THOR with IL and RL-finetuning methods, but the policy only generalizes to a small set of tasks due to the vision-based state representation. Using symbolic representations for state and action is also an inherent characteristic of works in task-and-motion-planning (Kaelbling and Lozano-Pérez, 2011; Konidaris et al., 2018) and symbolic planning (Asai and Fukunaga, 2017).

**World Models**: The concept of using TextWorld as a "game engine" to represent the world is broadly related to inverse graphics (Kulkarni et al., 2015) and inverse dynamics (Wu et al., 2017) where abstract visual or physical models are used for reasoning and future predictions. Similarly, some results in cognitive science suggest that humans use language as a cheaper alternative to sensorimotor simulation (Banks et al., 2020; Dove, 2014).

## 7 CONCLUSION

We introduced ALFWorld, the first interactive text environment with aligned embodied worlds. ALFWorld allows agents to explore, interact, and learn abstract polices in a textual environment. Pre-training our novel BUTLER agent in TextWorld, we show zero-shot generalization to embodied tasks in the ALFRED dataset. The results indicate that reasoning in textual space allows for better generalization to unseen tasks and also faster training, compared to other modalities like vision.

BUTLER is designed with modular components which can be upgraded in future work. Examples include the template-based state-estimator and the A* navigator which could be replaced with learned modules, enabling end-to-end training of the full pipeline. Another avenue of future work is to learn "textual dynamics models" through environment interactions, akin to vision-based world models (Ha and Schmidhuber, 2018). Such models would facilitate construction of text-engines for new domains, without requiring access to symbolic state descriptions like PDDL. Overall, we are excited by the challenges posed by aligned text and embodied environments for better cross-modal learning.

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

# A    DETAILS OF BUTLER::BRAIN

In this section, we use $o_t$ to denote text observation at game step $t$, $g$ to denote the goal description provided by a game.

We use $L$ to refer to a linear transformation and $L^f$ means it is followed by a non-linear activation function $f$. Brackets $[\cdot; \cdot]$ denote vector concatenation, $\odot$ denotes element-wise multiplication.

## A.1    OBSERVATION QUEUE

As mentioned in Section 3.1, we utilize an observation queue to cache the text observations that have been seen recently. Since the initial observation $o_0$ describes the high level layout of a room, including receptacles present in the current game, we it visible to BUTLER::BRAIN at all game steps, regardless of the length of the observation queue. Specifically, the observation queue has an extra space storing $o_0$, at any game step, we first concatenate all cached observations in the queue, then prepend the $o_0$ to form the input to the encoder. We find this helpful because it facilitates the pointer softmax mechanism in the decoder (described below) by guiding it to point to receptacle words in the observation. An ablation study on this is provided in Section 5.

## A.2    ENCODER

We use a transformer-based encoder, which consists of an embedding layer and a transformer block (Vaswani et al., 2017). Specifically, embeddings are initialized by pre-trained 768-dimensional BERT embeddings (Sanh et al., 2019). The embeddings are fixed during training in all settings.

The transformer block consists of a stack of 5 convolutional layers, a self-attention layer, and a 2-layer MLP with a ReLU non-linear activation function in between. In the block, each convolutional layer has 64 filters, each kernel's size is 5. In the self-attention layer, we use a block hidden size $H$ of 64, as well as a single head attention mechanism. Layernorm (Ba et al., 2016) is applied after each component inside the block. Following standard transformer training, we add positional encodings into each block's input.

At every game step $t$, we use the same encoder to process text observation $o_t$ and goal description $g$. The resulting representations are $h_{o_t} \in \mathbb{R}^{L_{o_t} \times H}$ and $h_g \in \mathbb{R}^{L_g \times H}$, where $L_{o_t}$ is the number of tokens in $o_t$, $L_g$ denotes the number of tokens in $g$, $H = 64$ is the hidden size.

## A.3    AGGREGATOR

We adopt the context-query attention mechanism from the question answering literature (Yu et al., 2018a) to aggregate the two representations $h_{o_t}$ and $h_g$.

Specifically, a tri-linear similarity function is used to compute the similarity between each token in $h_{o_t}$ with each token in $h_g$. The similarity between $i$-th token in $h_o$ and $j$-th token in $h_g$ is thus computed by (omitting game step $t$ for simplicity):

$$\text{Sim}(i, j) = W(h_{o_i}, h_{g_j}, h_{o_i} \odot h_{g_j}), \tag{1}$$

where $W$ is a trainable parameter in the tri-linear function. By applying the above computation for each $h_o$ and $h_g$ pair, we get a similarity matrix $S \in \mathbb{R}^{L_o \times L_g}$.

By computing the softmax of the similarity matrix $S$ along both dimensions (number of tokens in goal description $L_g$ and number of tokens in observation $L_o$), we get $S_g$ and $S_o$, respectively. The two representations are then aggregated by:

$$
\begin{aligned}
h_{og} &= [h_o; P; h_o \odot P; h_o \odot Q], \\
P &= S_g h_g^\top, \\
Q &= S_g S_o^\top h_o^\top,
\end{aligned}
\tag{2}
$$

where $h_{og} \in \mathbb{R}^{L_o \times 4H}$ is the aggregated observation representation.

Next, a linear transformation projects the aggregated representations to a space with size $H = 64$:

$$h_{og} = L^{\text{tanh}}(h_{og}). \tag{3}$$

To incorporate history, we use a recurrent neural network. Specifically, we use a GRU (Cho et al., 2014):

$$
\begin{aligned}
h_{\text{RNN}} &= \text{Mean}(h_{og}), \\
h_t &= \text{GRU}(h_{\text{RNN}}, h_{t-1}),
\end{aligned} \tag{4}
$$

in which, the mean pooling is performed along the dimension of number of tokens, i.e., $h_{\text{RNN}} \in \mathbb{R}^H$. $h_{t-1}$ is the output of the GRU cell at game step $t - 1$.

## A.4  DECODER

Our decoder consists of an embedding layer, a transformer block and a pointer softmax mechanism (Gulcehre et al., 2016). We first obtain the source representation by concatenating $h_{og}$ and $h_t$, resulting $h_{\text{src}} \in \mathbb{R}^{L_o \times 2H}$.

Similar to the encoder, the embedding layer is frozen after initializing it with pre-trained BERT embeddings. The transformer block consists of two attention layers and a 3-layer MLP with ReLU non-linear activation functions inbetween. The first attention layer computes the self attention of the input embeddings $h_{\text{self}}$ as a contextual encoding for the target tokens. The second attention layer then computes the attention $\alpha_{\text{src}}^i \in \mathbb{R}^{L_o}$ between the source representation $h_{\text{src}}$ and the $i$-th token in $h_{\text{self}}$. The $i$-th target token is consequently represented by the weighted sum of $h_{\text{src}}$, with the weights $\alpha_{\text{src}}^i$. This generates a source information-aware target representation $h_{\text{tgt}}' \in \mathbb{R}^{L_{\text{tgt}} \times H}$, where $L_{\text{tgt}}$ denotes the number of tokens in the target sequence. Next, $h_{\text{tgt}}'$ is fed into the 3-layer MLP with ReLU activation functions inbetween, resulting $h_{\text{tgt}} \in \mathbb{R}^{L_{\text{tgt}} \times H}$. The block hidden size of this transformer is $H = 64$.

Taking $h_{\text{tgt}}$ as input, a linear layer with tanh activation projects the target representation into the same space as the embeddings (with dimensionality of 768), then the pre-trained embedding matrix $E$ generates output logits (Press and Wolf, 2016), where the output size is same as the vocabulary size. The resulting logits are then normalized by a softmax to generate a probability distribution over all tokens in vocabulary:

$$p_a(y^i) = E^{\text{Softmax}}(L^{\text{tanh}}(h_{\text{tgt}})), \tag{5}$$

in which, $p_a(y^i)$ is the generation (abstractive) probability distribution.

We employ the pointer softmax (Gulcehre et al., 2016) mechanism to switch between generating a token $y^i$ (from a vocabulary) and pointing (to a token in the source text). Specifically, the pointer softmax module computes a scalar switch $s^i$ at each generation time-step $i$ and uses it to interpolate the abstractive distribution $p_a(y^i)$ over the vocabulary (Equation 5) and the extractive distribution $p_x(y^i) = \alpha_{\text{src}}^i$ over the source text tokens:

$$p(y^i) = s^i \cdot p_a(y^i) + (1 - s^i) \cdot p_x(y^i), \tag{6}$$

where $s^i$ is conditioned on both the attention-weighted source representation $\sum_j \alpha_{\text{src}}^{i,j} \cdot h_{\text{src}}^j$ and the decoder state $h_{\text{tgt}}^i$:

$$s^i = L_1^{\text{sigmoid}}(\tanh(L_2(\sum_j \alpha_{\text{src}}^{i,j} \cdot h_{\text{src}}^j) + L_3(h_{\text{tgt}}^i))). \tag{7}$$

In which, $L_1 \in \mathbb{R}^{H \times 1}$, $L_2 \in \mathbb{R}^{2H \times H}$ and $L_3 \in \mathbb{R}^{H \times H}$ are linear layers, $H = 64$.

## B  TRAINING AND IMPLEMENTATION DETAILS

In this section, we provide hyperparameters and other implementation details.

For all experiments, we use *Adam* (Kingma and Ba, 2014) as the optimizer. The learning rate is set to 0.001 with a clip gradient norm of 5.

During training with DAgger, we use a batch size of 10 to collect transitions (tuples of $\{o_0, o_t, g, \hat{a}_t\}$) at each game step $t$, where $\hat{a}_t$ is the ground-truth action provided by the rule-based expert (see Section E). We gather a sequence of transitions from each game episode, and push each sequence into a replay buffer, which has a capacity of 500K episodes. We set the max number of steps per episode to be 50. If the agent uses up this budget, the game episode is forced to terminate. We linearly anneal the fraction of the expert's assistance from 100% to 1% across a window of 50K episodes.

The agent is updated after every 5 steps of data collection. We sample a batch of 64 data points from the replay buffer. In the setting with the recurrent aggregator, every sampled data point is a sequence of 4 consecutive transitions. Following the training strategy used in the recurrent DQN literature (Hausknecht and Stone, 2015; Yuan et al., 2018), we use the first 2 transitions to estimate the recurrent states, and the last 2 transitions for updating the model parameters.

BUTLER::BRAIN learns to generate actions token-by-token, where we set the max token length to be 20. The decoder stops generation either when it generates a special end-of-sentence token [EOS], or hits the token length limit.

When using the beam search heuristic to recover from failed actions (see Figure 5), we use a beam width of 10, and take the top-5 ranked outputs as candidates. We iterate through the candidates in the rank order until one of them succeeds. This heuristic is not always guaranteed to succeed, however, we find it helpful in most cases. Note that we do not employ beam search when we evaluate during the training process for efficiency, e.g., in the seen and unseen curves shown in Figure 4. We take the best performing checkpoints and then apply this heuristic during evaluation and report the resulting scores in tables (e.g., Table 2).

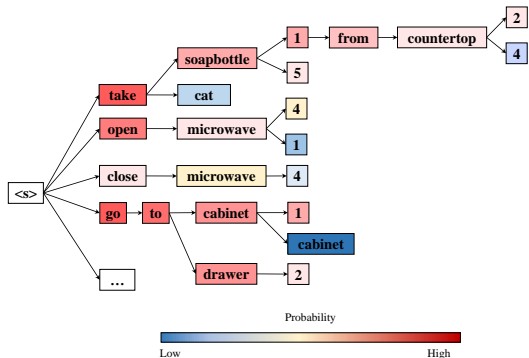

Figure 5: Beam search for recovery actions.

By default unless mentioned otherwise (ablations), we use all available training games in each of the task types. We use an observation queue length of 5 and use a recurrent aggregator. The model is trained with DAgger, and during evaluation, we apply the beam search heuristic to produce the reported scores. All experiment settings in TextWorld are run with 8 random seeds. All text agents are trained for 50,000 episodes.

## C  TEXTWORLD ENGINE

Internally, the TextWorld Engine is divided into two main components: a planner and text generator.

**Planner**  TextWorld Engine uses Fast Downward (Helmert, 2006), a domain-independent classical planning system to maintain and update the current state of the game. A state is represented by a set of predicates which define the relations between the entities (objects, player, room, etc.) present in the game. A state can be modified by applying production rules corresponding to the actions listed in Table 6. All variables, predicates, and rules are defined using the PDDL language.

For instance, here is a simple state representing a player standing next to a microwave which is closed and contains a mug:

$$s_t = \texttt{at}(player, microwave) \otimes \texttt{in}(mug, microwave)$$
$$\otimes \texttt{closed}(microwave) \otimes \texttt{openable}(microwave),$$

where the symbol $\otimes$ is the linear logic *multiplicative conjunction* operator. Given that state, a valid action could be **open** $microwave$, which would essentially transform the state by replacing $\texttt{closed}(microwave)$ with $\texttt{open}(microwave)$.

**Text generator**  The other component of the TextWorld Engine, the text generator, uses a context-sensitive grammar designed for the ALFRED environments. The grammar consists of text templates similar to those listed in Table 6. When needed, the engine will sample a template given some context,

i.e., the current state and the last action. Then, the template gets realized using the predicates found in the current state.

## D  MASK R-CNN DETECTOR

We use a Mask R-CNN detector (He et al., 2017) pre-trained on MSCOCO (Lin et al., 2014) and fine-tune it with additional labels from ALFRED training scenes. To generate additional labels, we replay the expert demonstrations from ALFRED and record ground-truth image and instance segmentation pairs from the simulator (THOR) after completing each high-level action e.g., goto, pickup etc. We generate a dataset of 50K images, and fine-tune the detector for 4 epochs with a batch size of 8 and a learning rate of $5e$-$4$. The detector recognizes 73 object classes where each class could vary up to 1-10 instances. Since demonstrations in the kitchen are often longer as they involve complex sequences like heating, cleaning etc., the labels are slightly skewed towards kitchen objects. To counter this, we balance the number of images sampled from each room (kitchen, bedroom, livingroom, bathroom) so the distribution of object categories is uniform across the dataset.

## E  RULE-BASED EXPERT

To train text agents in an imitation learning (IL) setting, we use a rule-based expert for supervision. A given task is decomposed into sequence of subgoals (e.g., for heat & place: find the object, pick the object, find the microwave, heat the object with the microwave, find the receptacle, place the object in the receptacle), and a closed-loop controller tries to sequentially execute these goals. We note that while designing rule-based experts for ALFWorld is relatively straightforward, experts operating directly in embodied settings like the PDDL planner used in ALFRED are prone to failures due to physical infeasibilities and non-deterministic behavior in physics-based environments.

## F  BENEFITS OF TRAINING IN TEXTWORLD OVER EMBODIED WORLD

Pre-training in TextWorld offers several benefits over directly training in embodied environments. Figure 6 presents the performance of an expert (that agents are trained to imitate) across various environments. The abstract textual space leads to higher goal success rates resulting from successful navigation and manipulation subroutines. TextWorld agents also do not suffer from object mis-detections and slow execution speed.

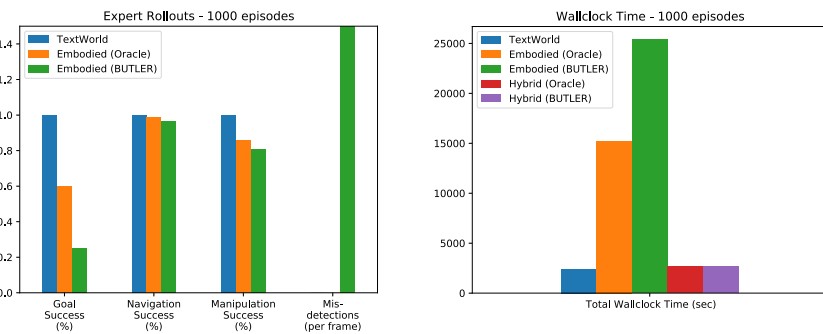

Figure 6: **Domain Analysis:** The performance of an expert across various environments.

# G  OBSERVATION TEMPLATES

The following templates are used by the state-estimator to generate textual observations $o_t$. The object IDs **{obj id}** correspond to Mask R-CNN objects detection or ground-truth instance IDs. The receptacle IDs **{recep id}** are based on the receptacles listed in the initial observation $o_0$. Failed actions and actions without any state-changes result in `Nothing happens`.

| Actions | Templates |
|---|---|
| **goto** | (a) `You arrive at `**{loc id}**`.  On the `**{recep id}**`,`
`   you see a `**{obj1 id}**`, ...  and a `**{objN id}**`.`
(b) `You arrive at `**{loc id}**`.  The `**{recep id}**` is closed.`
(c) `You arrive at `**{loc id}**`.  The `**{recep id}**` is open.`
`   On it, you see a `**{obj1 id}**`, ...  and a `**{objN id}**`.` |
| **take** | `You pick up the `**{obj id}**` from the `**{recep id}**`.` |
| **put** | `You put the `**{obj id}**` on the `**{recep id}**`.` |
| **open** | (a) `You open the `**{recep id}**`.  In it,`
`   you see a `**{obj1 id}**`, ...  and a `**{objN id}**`.`
(b) `You open the `**{recep id}**`.  The `**{recep id}**` is empty.` |
| **close** | `You close the `**{recep id}**`.` |
| **toggle** | `You turn the `**{obj id}**` on.` |
| **heat** | `You heat the `**{obj id}**` with the `**{recep id}**`.` |
| **cool** | `You cool the `**{obj id}**` with the `**{recep id}**`.` |
| **clean** | `You clean the `**{obj id}**` with the `**{recep id}**`.` |
| **inventory** | (a) `You are carrying: `**{obj id}**`.`
(b) `You are not carrying anything.` |
| **examine** | (a) `On the `**{recep id}**`, you see a `**{obj1 id}**`, ...`
`   and a `**{objN id}**`.`
(b) `This is a hot/cold/clean `**{obj}**`.` |

Table 6: High-level text actions supported in ALFWorld along with their observation templates.

# H  Goal Descriptions

## H.1  Templated Goals

The goal instructions for training games are generated with following templates. Here `obj`, `recep`, `lamp` refer to object, receptacle, and lamp classes, respectively, that pertain to a particular task. For each task, the two corresponding templates are sampled with equal probability.

| task-type | Templates |
|---|---|
| Pick & Place | (a) `put a {obj} in {recep}.`
(b) `put some {obj} on {recep}.` |
| Examine in Light | (a) `look at {obj} under the {lamp}.`
(b) `examine the {obj} with the {lamp}.` |
| Clean & Place | (a) `put a clean {obj} in {recep}.`
(b) `clean some {obj} and put it in {recep}.` |
| Heat & Place | (a) `put a hot {obj} in {recep}.`
(b) `heat some {obj} and put it in {recep}.` |
| Cool & Place | (a) `put a cool {obj} in {recep}.`
(b) `cool some {obj} and put it in {recep}.` |
| Pick Two & Place | (a) `put two {obj} in {recep}.`
(b) `find two {obj} and put them {recep}.` |

Table 7: Task-types and the corresponding goal description templates.

## H.2  Human Annotated Goals

The human goal descriptions used during evaluation contain 66 unseen verbs and 189 unseen nouns with respect to the templated goal instructions used during training.

**Unseen Verbs:**   acquire, arrange, can, carry, chill, choose, cleaning, clear, cook, cooked, cooled, dispose, done, drop, end, fill, filled, frying, garbage, gather, go, grab, handled, heated, heating, hold, holding, inspect, knock, left, lit, lock, microwave, microwaved, move, moving, pick, picking, place, placed, placing, putting, read, relocate, remove, retrieve, return, rinse, serve, set, soak, stand, standing, store, take, taken, throw, transfer, turn, turning, use, using, walk, warm, wash, washed.

**Unseen Nouns:**   alarm, area, back, baisin, bar, bars, base, basin, bathroom, beat, bed, bedroom, bedside, bench, bin, books, bottle, bottles, bottom, box, boxes, bureau, burner, butter, can, canteen, card, cardboard, cards, cars, cds, cell, chair, chcair, chest, chill, cistern, cleaning, clock, clocks, coffee, container, containers, control, controllers, controls, cooker, corner, couch, count, counter, cover, cream, credit, cupboard, dining, disc, discs, dishwasher, disks, dispenser, door, drawers, dresser, edge, end, floor, food, foot, freezer, game, garbage, gas, glass, glasses, gold, grey, hand, head, holder, ice, inside, island, item, items, jars, keys, kitchen, knifes, knives, laddle, lamp, lap, left, lid, light, loaf, location, lotion, machine, magazine, maker, math, metal, microwaves, move, nail, newsletters, newspapers, night, nightstand, object, ottoman, oven, pans, paper, papers, pepper, phone, piece, pieces, pillows, place, polish, pot, pullout, pump, rack, rag, recycling, refrigerator, remote, remotes, right, rinse, roll, rolls, room, safe, salt, scoop, seat, sets, shaker, shakers, shelves, side, sink, sinks, skillet, soap, soaps, sofa, space, spatulas, sponge, spoon, spot, spout, spray, stand, stool, stove, supplies, table, tale, tank, television, textbooks, time, tissue, tissues, toaster, top, towel, trash, tray, tv, vanity, vases, vault, vegetable, wall, wash, washcloth, watches, water, window, wine.

# I  Action Candidates vs Action Generation

BUTLER::BRAIN generates actions in a token-by-token fashion. Prior text-based agents typically use a list of candidate commands from the game engine (Adhikari et al., 2020) or populate a list of command templates (Ammanabrolu and Hausknecht, 2020). We initially trained our agents with candidate commands from the TextWorld Engine, but they quickly ovefit without learning affordances,

commonsense, or pre-conditions, and had zero performance on embodied transfer. In the embodied setting, without access to a TextWorld Engine, it is difficult to generate candidate actions unless a set of heuristics is handcrafted with strong priors and commonsense knowledge. We also experimented with populating a list of command templates, but found this to be infeasible as some scenarios involved 1000s of populated actions per game step.

## J  ALFRED TASK DESCRIPTIONS

The following descriptions describe the processes involved in each of six `task-types`:

- Pick & Place (e.g., "put a plate on the coffee table") - the agent must find an object of the desired type, pick it up, find the correct location to place it, and put it down there.
- Examine in Light (e.g., "examine a book under the lamp") - the agent must find an object of the desired type, locate and turn on a light source with the desired object in-hand.
- Clean & Place (e.g., "clean the knife and put in the drawer") - the agent must find an object of the desired type, pick it up, go to a sink or a basin, wash the object by turning on the faucet, then find the correct location to place it, and put it down there.
- Heat & Place (e.g., "heat a mug and put on the coffee table") - the agent must find an object of the desired type, pick it up, go to a microwave, heat the object turning on the microwave, then find the correct location to place it, and put it down there.
- Cool & Place (e.g., "put a cool bottle on the countertop") - the agent must find an object of the desired type, pick it up, go to a fridge, put the object inside the fridge and cool it, then find the correct location to place it, and put it down there.
- Pick Two & Place (e.g., "put two pencils in the drawer") - the agent must find an object of the desired type, pick it up, find the correct location to place it, put it down there, then look for another object of the desired type, pick it up, return to previous location, and put it down there with the other object.

# K ALFWORLD TEXT GAME EXAMPLES

Three examples of ALFWorld text game walkthroughs:

```
You are in the middle of a room. Looking quickly around you, you see a towelholder 1,
a toilet 1, a bathtubbasin 1, a drawer 4, a handtowelholder 2, a drawer 6, a drawer 1,
a countertop 1, a sinkbasin 1, a drawer 2, a drawer 3, a toiletpaperhanger 1, a drawer 5,
a handtowelholder 1, a towelholder 2, a sinkbasin 2, and a garbagecan 1.

Your task is to: put a clean cloth in bathtubbasin.

> go to drawer 5
You arrive at loc 9. The drawer 5 is closed.
> open drawer 5
You open the drawer 5. The drawer 5 is open. In it, you see a candle 1.
> close drawer 5
You close the drawer 5.
> go to countertop 1
You arrive at loc 25. On the countertop 1, you see a cloth 2, a soapbottle 1, a cloth 1,
and a candle 3.
> take cloth 1 from countertop 1
You pick up the cloth 1 from the countertop 1.
> go to sinkbasin 1
You arrive at loc 20. On the sinkbasin 1, you see nothing.
> clean cloth 1 with sinkbasin 1
You clean the cloth 1 using the sinkbasin 1.
> go to bathtubbasin 1
You arrive at loc 14. On the bathtubbasin 1, you see a soapbar 1.
> put cloth 1 in/on bathtubbasin 1
You won!
▬ ▬ ▬ ▬ ▬ ▬ ▬ ▬ ▬ ▬ ▬ ▬ ▬ ▬ ▬ ▬ ▬ ▬ ▬ ▬ ▬ ▬ ▬ ▬ ▬ ▬ ▬ ▬ ▬ ▬ ▬ ▬ ▬ ▬ ▬ ▬ ▬ ▬ ▬ ▬ ▬
You are in the middle of a room. Looking quickly around you, you see a drawer 15, a drawer 19,
a drawer 7, a armchair 1, a drawer 3, a drawer 16, a drawer 2, a drawer 17, a sidetable 3,
a sidetable 5, a drawer 4, a drawer 20, a sofa 1, a drawer 13, a drawer 12, a drawer 9,
a drawer 21, a sidetable 4, a sidetable 2, a drawer 5, a drawer 11, a drawer 1, a garbagecan 1,
a dresser 1, a cabinet 1, a drawer 8, a drawer 18, a sidetable 1, a drawer 14, a drawer 6,
and a drawer 10.

Your task is to: find two remotecontrol and put them in armchair.

> go to sidetable 2
You arrive at loc 34. On the sidetable 2, you see a remotecontrol 1.
> take remotecontrol 1 from sidetable 2
You pick up the remotecontrol 1 from the sidetable 2.
> go to armchair 1
You arrive at loc 1. On the armchair 1, you see nothing.
> put remotecontrol 1 in/on armchair 1
You put the remotecontrol 1 in/on the armchair 1.
> go to dresser 1
You arrive at loc 18. On the dresser 1, you see a creditcard 1, a television 1, a laptop 1,
a book 1, a statue 2, and a cellphone 2.
> go to sidetable 5
You arrive at loc 17. On the sidetable 5, you see a box 1, and a creditcard 2.
> go to sofa 1
You arrive at loc 2. On the sofa 1, you see a newspaper 1, a pillow 1, and a remotecontrol 2.
> take remotecontrol 2 from sofa 1
You pick up the remotecontrol 2 from the sofa 1.
> go to armchair 1
You arrive at loc 1. On the armchair 1, you see a remotecontrol 1.
> put remotecontrol 2 in/on armchair 1
You won!

▬ ▬ ▬ ▬ ▬ ▬ ▬ ▬ ▬ ▬ ▬ ▬ ▬ ▬ ▬ ▬ ▬ ▬ ▬ ▬ ▬ ▬ ▬ ▬ ▬ ▬ ▬ ▬ ▬ ▬ ▬ ▬ ▬ ▬ ▬ ▬ ▬ ▬ ▬ ▬ ▬
You are in the middle of a room. Looking quickly around you, you see a safe 1, a shelf 4, a drawer 2,
a bed 1, a drawer 1, a shelf 5, a shelf 2, a sidetable 2, a shelf 3, a drawer 3, a shelf 1, a sidetable 1,
a desk 1, and a garbagecan 1.

Your task is to: examine the alarmclock with the desklamp.

> go to desk 1
You arrive at loc 8. On the desk 1, you see a pen 1, a bowl 1, a alarmclock 2, a pencil 2, a pencil 3,
a creditcard 3, a book 1, a alarmclock 3, a keychain 3, and a book 2.
> take alarmclock 2 from desk 1
You pick up the alarmclock 2 from the desk 1.
> go to sidetable 2
You arrive at loc 1. On the sidetable 2, you see a desklamp 1, and a alarmclock 1.
> use desklamp 1
You won!
```

