# OpenReview forum: "ALFWorld: Aligning Text and Embodied Environments for Interactive Learning"
_ICLR.cc/2021/Conference — ICLR 2021 Poster_

### Official Review · AnonReviewer1 · 2020-10-27
**Lacks clarity on the usefulness of the proposed ALFWorld environment**

**Rating:** 4
**Confidence:** 5

**Review:**

This work suggests an approach to improve the embodied agents interacting in an environment solving language-specified tasks. The approach introduces a parallel text-based (minus any physical interaction or visual input) environment which can be used to pretrain the agent by learning language-only policies. This text-based environment is an extension of the TextWorld framework.

Strengths:

* The paper is clearly written.
* The premise of the paper that such a parallel environment allows agents to explore/learn in an abstract language-only environment while also allowing transfer of the learned knowledge to physical embodied environments is well motivated.

Weaknesses:

* The main premise is that such a text-based environment can be used to learn abstract (high-level) policies that can then be transferred to an embodied agent to solve language-specified tasks in a physically simulated (with visual input) environment. However, the main experiment to prove this claim in Table 4 falls short of proving this. The results in that table show agents learned with an oracle state estimator which means there is no visual input processing during this mode. It can also be noted that the Controller is also a heuristic module with no learning. Which means under this setting, learning/evaluating in embodied environment is pure text-driven which is same as TextWorld, hence it is not surprising that TW-Only performs better.
* The performance of BUTLER-ORACLE in Table 3 is similar to the performance of BUTLER on TextWorld (on All Tasks) which further proves that using oracle state estimator is essentially reducing the embodied environment tasks to TextWorld task.
* Furthermore, given that this is the main premise of the paper, Table 4 needs to be complete with all the tasks.
* It is unclear why the components in BUTLER Agent interact using discretized constructs (of natural language) instead of continuous vector spaces.
* There seem to be several hacks adopted to obtain good results (such as using beam search during evaluation, using rule-based expert for supervision) which make this approach highly specific to the given dataset/environment limiting its general applicability.

Overall: The strong motivation of this work is not supported by empirical results. The results presented in the paper use a number of hacks, are incomplete and lack insights.

Update after author response:

The author response is much appreciated. However, my two main concerns remain unaddressed. The authors may add these additional experiments/results to Table 3 and 4 in further revisions for a stronger submission.
* Table 3 is the main result of the paper which claims policies learned in TextWorld (TW) environment can be transferred over to ALFRED (ALF) environment under zero-shot setting. This result alone has several weaknesses -- (1) Evaluation done on non-human goals in ALF seem to use same template as that used in TW, so it is not surprising that agents will have non-zero success rates on similar language specifications. (2) When evaluation is done on human goals (which seems to be the real test), the agent's performance is very low. Also, there are no baselines (e.g., random) provided to compare those scores against. (3) Why are the experiments only conducted in zero-shot setting? This actually brings me to the second weakness.

* Since the transfer learning is happening from a pure-text TW environment to a physically simulated (with visual input) ALF environment, it is more interesting/relevant to see how the language module pre-trained on text-only TW adapts to multimodal setup in ALF. This adaptation will require further training/fine-tuning on ALF so that visual/control modules can adapt to this pre-trained language module. This experiment was attempted in Table 4, however, as pointed in my initial review, this falls short of proving any claims made in the paper because the agents in Table 4 are learned with an oracle state estimator which means there is no visual input processing during this mode. It can also be noted that the Controller is also a heuristic module with no learning. Which means the setting used in Table 4 reduces learning/evaluating in embodied ALF environment to a pure text-driven environment.

---

> ### Author Response · Authors · 2020-11-16
> **Response to R1 - Some clarifications and justifications (Part 1)**
>
> We thank R1 for providing detailed feedback on the paper. We are glad that R1 appreciates the motivation behind pre-training agents in abstract language-based environments.
>
>
> – Re: Weakness 1 & 2
> We would like to clarify some misunderstandings. We note that Table 4 is not the main result of the paper, but rather an ablation study to investigate various training strategies. The main result is presented in Table 3, which supports our claim that high-level language polices can be transferred to physically embodied environments. In particular, BUTLER, our main agent, transfers to the embodied setting in a zero-shot manner while using a (non-oracle) Mask-RCNN to perform state estimation. The BUTLER-Oracle agent which does use oracle state estimation serves as an upper bound to illustrate the possible performance of the system given perfect object detection, manipulation, and navigation.
>
> In contrast, results in Table 4 are intended to answer the auxiliary question of "how efficient is it to train in TextWorld, and then transfer to the embodied setting versus simply training from scratch in the embodied world?" To address this question, we chose to use BUTLER-Oracle (oracle state estimator + oracle controller) since it would give the Embodied-only agent the best possible performance. Moreover, we agree that the controller and pre-trained state-estimator are not being learned during the course of embodied interaction, and in future work it may be possible to get better performance with a fully-learned end-to-end pipeline. To this end, a benefit of BUTLER’s modular design is the ability to easily "swap in" learned components to replace parts of this pipeline.
>
> Lastly, with respect to “using oracle state estimator is essentially reducing the embodied environment tasks to TextWorld task”, the difference between TW-Only and Embodied-Only successes in Table 4 indicates the opposite. This difference is a result of the discrepancies between the two domains: occluded objects, physical restrictions, execution failures, and execution speed. In Table 3, the TextWorld and BUTLER-Oracle numbers are not directly comparable, since the former uses TextWorld evaluations (no physical interactions), and the latter uses embodied evaluations (physical interactions), while all strategies in Table 4 use embodied evaluations.
>
> We thank R1 for highlighting these points. Certain nuances in the results section are hard to discern. We tried to make these differences clearer in the writing.
>
>
> – Re: Weakness 3
> We will report training strategy ablations for the All Tasks setting. Again, we would like to emphasize that the main TextWorld transfer results are presented in Table 3, not Table 4. We do not present full ablation results for each task-type in Table 4 as these experiments are resource intensive. Each row in Table 4 that involves embodied training takes 3-6 days on a single GPU+CPU instance due to the speed of the embodied simulator (AI2THOR).
>
>
> – Re: Weakness 4
> In designing the overall system, we started with the hypothesis that reasoning in space of high-level language would permit generalization to unseen environments. BULTER::Brain is designed to interface with the world through sequences of discrete words. Arguably the most natural way to integrate this text-agent into the larger embodied BUTLER agent is to respect the language based interface used by BUTLER::Brain.
>
> Further, note that the MaskRCNN-FCN baseline in Table 5 uses continuous features from BUTLER’s pre-trained detector in lieu of discrete symbols constructed from object classes. As indicated by MaskRCNN-FCN’s poor performance and as also noted by prior works (Zhang et. al 2020, Shridhar et. al 2020), continuous visual features tend to overfit to training scenes and generalize poorly to unseen environments.
>
> Additionally, using language as the intermediate state and action representation makes the agent’s decisions interpretable, allowing for easier integration into human-robot collaborative systems.

---

> > ### Author Response · Authors · 2020-11-16
> > **Response to R1 - Some clarifications and justifications (Part 2)**
> >
> > – Re: Weakness 5
> > Beam search is a standard technique used in natural language generation (NLG) works such as dialogue generation (Kool et al. 2019), image captioning (Vijayakumar et al. 2018), summarization (Gehrmann et al. 2018) and keyphrase generation (Meng et al. 2018). Beam search is typically used in tasks that favor higher recall. For instance, in standard key phrase generation setting, models apply beam search and take generated sequences from all beams as a set of outputs. Also, note that we use beam-search simply to improve text-generation, and not to optimize the performance of embodied interactions. Our action-generation module is quite generic, and is applicable to a wide range of text-based interaction problems since we do not rely on any domain-specific command templates.
> >
> > We agree that the rule-based expert is a strong assumption. As noted in Appendix I, we experimented with RL methods that didn’t rely on this expert, but they failed to successfully generalize between the 100s to 1000s of training tasks. In our code-release, we provide a state-of-the-art TextDQN agent for the community to experiment with. We are excited to see if ALFWorld tasks are indeed solvable without expert demonstrations.
> >
> > Overall, we note that BUTLER is modular framework where each component – vision, control, high-level reasoning, can be improved upon independently. Modular frameworks have shown to be applicable to a variety of vision-language
> > grounding problems (Hu et. al 2018, Andreas et. al 2017, Das et. al 2018).
> >
> > References:
> > Zhang et.al 2020, "Diagnosing the Environment Bias in Vision-and-Language Navigation", https://arxiv.org/abs/2005.03086
> > Shridhar et. al 2020, "ALFRED: A benchmark for interpreting grounded instructions for everyday tasks", https://arxiv.org/abs/1912.01734
> > Kool et. al 2019, "Stochastic Beams and Where to Find Them: The Gumbel-Top-k Trick for Sampling Sequences Without Replacement", https://arxiv.org/abs/1903.06059
> > Vijayakumar et. al 2018, "Diverse Beam Search: Decoding Diverse Solutions from Neural Sequence Models", https://arxiv.org/abs/1903.06059
> > Gehrmann et. al 2018, "Bottom-Up Abstractive Summarization", https://arxiv.org/abs/1808.10792
> > Meng et. al 2018, "Deep Keyphrase Generation", https://arxiv.org/abs/1704.06879
> > Hu et. al 2017, "Learning to Reason: End-to-End Module Networks for Visual Question Answering", https://arxiv.org/abs/1704.05526
> > Andreas et. al 2016, "Neural Modular Networks", https://arxiv.org/abs/1511.02799
> > Das et. al 2018, "Neural Modular Control for Embodied Question Answering", https://arxiv.org/abs/1810.11181

---

> ### Author Response · Authors · 2020-11-23
> **R1 Follow-up**
>
> Following up on this, we wanted to check if our response and updates to the paper addressed your concerns. We'd be happy to provide more details, so please let us know if you have additional questions or if there is anything we can do to help.

---

### Official Review · AnonReviewer3 · 2020-10-28
**Recommendation to Accept**

**Rating:** 6
**Confidence:** 4

**Review:**

Summary:

This paper describes an approach for training an agent which completes goals specified via a language instruction in the ALFRED environment. The paper proposes a model and training scheme.

The model decomposes the problem into three steps: (1) a perceptual module which takes as input an environment observation and generates a textual description of it including the objects and their spatial relations, (2) a goal-planning module which takes as input the high-level goal (which may require completing multiple subgoals), the textual description generated from module 1 and generates a textual description of a subgoal, such as an action the agent should take with arguments, and (3) a controller module which takes as input the state of the environment and the subgoal description generated from module 2, and generates a sequence of actions which execute this subgoal. The entire model is ran each time the agent completes a subgoal, using the current environment observation as input to module 1.

The proposed training scheme focuses on pre-training the second module. This module is pre-trained using a proposed environment, ALFWorld, which is modeled after text adventure games like TextWorld (Côté et al. 2018), and is intended to abstract away the perception and control problems from the reasoning module. Training environments in ALFRED are converted into ALFWorld environments, and for each training goal, an oracle is constructed (which is a function that maps from any ALFWorld environment state to an optimal subsequent action that leads to the goal). This facilitates pre-training the second module using DAgger through interactions with ALFWorld.

----------------------------------

Reasons for score:

I vote for accepting the paper. The idea of explicitly training the model to reason about subgoals is intriguing. However, there are several assumptions and limitations of the proposed approach.

----------------------------------

Strengths:

- A model which decomposes high-level goals into low-level action sequences is very valuable and interesting.
- Comparison of single-goal vs. multiple-goal models.
- Good set of ablations, experiments, and comparisons, although I am not sure why Section 4.3 only looked at a single task.

----------------------------------

Weaknesses:

- Because ALFWorld is an abstraction of the problem, it must make assumptions about perception and/or physical control. The paper describes several simplifications it makes: e.g., as far as I can tell, it disregards physical attributes of objects, and only retains the existence of particular object types when BUTLER::Vision processes an observation. Also, ALFWorld does not implement physical/semantic constraints (e.g., what can fit in a microwave). The setup of BUTLER means that losing perceived attributes in the output of the first module may result in the reasoning module being unable to reason about these attributes, and it is up to the designers of ALFWorld to decide which attributes are important to retain.
- As above, ALFWorld is manually designed and requires making decisions about what aspects of the problem to model and which aspects should be abstracted away. This also includes requiring design of an oracle function, or alternatively an effective reward function (i.e., a denser reward function than binary task completion), which was not explored.

----------------------------------

Questions for the authors:

- Can you elaborate more on what it means for a sequence to be prototypical (in the abstract)?
- What is the difference between the high-level goals and low-level step-by-step language in ALFRED? Is this similar to the goal "put a pan on the dining table" vs. "go to the cabinet", "open the cabinet"?
- What exactly are the generalization dimensions for unseen rooms? Novel combinations of object instances that existed during training, novel instances (e.g., color) of object types that existed during training, or completely novel object types? Or something else?
- In the last paragraph of section 3.1, does this refer to playing a game at inference time using ALFRED, or during training with ALFWorld?
- Beam search is only done on the subgoal descriptions output by BUTLER::Brain, not the action actions taken in the ALFRED environment by the controller?
- If the Human Goals setting is using human-written (natural language) goal specifications, then what are the goals in the other evaluation settings? Auto-generated?

----------------------------------

Additional feedback:

- The color coding that begins on the third page is confusing, and the colors are very hard to tell apart (at least for me).
- In the related work on embodied language learning, the last sentence is a bit confusing as ALFRED is a fully interactive modality, right? If not, what is considered fully-interactive?

----------------------------------

Rating after discussion: lowering to a 6, as I share concerns with R1 about experiments and generalizability of the proposed approach.

---

> ### Author Response · Authors · 2020-11-16
> **Response to R3 - Thanks for the positive response. We have updated the paper to address your comments.**
>
> We thank R3 for the detailed and insightful feedback. We are glad R3 appreciates our approach to reasoning about subgoals in an abstract space.
>
> – Re: Strength 3
> We will update Table 4 to report successes for the All Tasks setting.
>
> – Re: Weakness 1
> We agree that the text-based environment is limited by the things it abstracts away about the embodied world. But we believe some level of abstraction is necessary for embodied tasks. Imagine cooking a new dish in a new kitchen. It’s unlikely that you have a good physical model of every physical implement, e.g. sliding drawers, cutting with knives, etc. But you can still reason abstractly about the sequence of steps needed to be taken without worrying about the physical and visual details involved in executing those steps.
>
> However, as R3 mentions, some attributes of objects could give useful clues about how to use or interact with novel or unseen objects. Although ALFWorld doesn’t currently expose physical or detailed visual attributes, the framework is extensible and could be modified in this direction by future work.
>
> – Re: Weakness 2
> The BUTLER agent is trained with imitation learning from expert demonstrations, which does not rely on any ‘reward’ mechanism. But we agree with R3 that dense-reward based RL could be an interesting direction for future works.
>
> – Question 1
> Here “prototypicality” refers to the canonical sequence of subgoals corresponding to a task, e.g: “heat X and put it in Y” implies ‘find X’, ‘pick X’, ‘find microwave’, and so on.
>
> – Question 2
> The high-level goal is an ambiguous description of what needs to be achieved without explaining how it can be achieved. The low-level step-by-step instructions guide the agent on how to actually achieve a goal, for e.g: “turn right and walk straight to the refrigerator”, “open the fridge and pick the apple”. We updated Section 2 Paragraph 2 to reflect this.
>
> – Question 3
> We describe this in Section 2, Paragraph 3, Point 2. The unseen tasks contain seen and unseen object-receptacle pairs but all set in unseen rooms with vary- ing visual features like texture, color etc. For the full distribution of object categories we refer to Figure F6 in the ALFRED paper (Shridhar et.al 2020).
>
> – Question 4
> It corresponds to ‘inference time’ in both TextWorld (Table 2) and the embodied world (Table 3). We updated Section 3.1, Paragraph 2.
>
> – Question 5
> Correct, beam-search is simply used to improve the generated action sentence, not to optimize over the low-level embodied actions. We updated Section 3.1, Paragraph 2.
>
> – Question 6
> Yes, unless specified, we use templated goals. We updated Section 3.1, Paragraph 1.
>
> – Additional Feedback 1
> We updated some of the color codings.
>
> – Additional Feedback 2
> ALFRED is a fully-interactive embodied modality, but it does not contain a fully-interactive textual modality. The textual modality refers to an interactive system that generates textual observations and executes textual commands without relying on an embodied simulator. We updated the corresponding related work section.

---

### Official Review · AnonReviewer2 · 2020-10-28
**An Aligned TextWorld and Visual Environment**

**Rating:** 7
**Confidence:** 3

**Review:**

The paper presents a new interactive environment which is both text-based and contains visual simulation which are aligned. The authors also propose a first agent architecture which uses the visual observations as well as the text-based (named BUTLER). The authors tested the generalization capabilities of the proposed BUTLER architecture compared to a seq2seq transformer model.

Strong points:
- novel environment for text-based and aligned visual content (could potentially lead to follow up research) - a significant contribution to the community.
- have demonstrated that visual representation helps to generalize in these kind of environments (text + visual)
- the paper is nicely written and easy to follow
- the figures plots and tables are clear and help to understand the research

Weak points:
- the complex system: e.g.,  a pre-trained M-RCNN, a pre-trained text-agent, and training the text-agent using imitation learning (DAgger) biases the experimentation (makes the results less convincing). Maybe an intermediate naive baseline should have been considered.

---

> ### Author Response · Authors · 2020-11-16
> **Response to R2 - Thanks for the positive response.**
>
> We are glad R2 finds ALFWorld a significant contribution to the community.
>
> – Re: Weakness 1
>
> We agree with R2 that the current system is somewhat complex, which is due in a large part to the challenge of trying to solve 100s to 1000s of unique training scenes and generalizing to new test tasks.
>
> As we noted in Appendix I, we found that current RL agents simply are not able to generalize to the diversity and magnitude of tasks presented by ALFWorld. Our code-release provides an RL-based TextDQN agent for the community to experiment with or build on.
>
> Moreover, rather than reducing the complexity of ALFWorld to enable current RL agents, we chose to keep the domain challenging and explore alternatives such as DAgger to make learning feasible. Broadly, we believe that in building a compelling benchmark for the community, it makes sense to propose something outside of the reach of current methods such that future work can leverage advances to remove or replace the DAgger training and relax some of the pre-trained components (such as Mask-RCNN). To enable this, we have designed BUTLER as a modular system to facilitate extending or replacing components in the pipeline.

---

### Author Response · Authors · 2020-11-16
**General Updates (Nov 16)**

We thank all the reviewers for their insightful and constructive feedback. To address R1 and R3’s comment on reporting All Tasks results for Table 4, we are currently running these experiments, and we will update the paper as soon as the results are ready. Other updates to the paper are summarized below:

1. Updated Figure 4 with All Tasks results for consistency.
2. Addressed the detail questions from R3.
3. Added related work from task-and-motion-planning and symbolic planning literature.
4. Updated the title to reflect the broader theme of the paper.

---

### Author Response · Authors · 2020-11-23
**General Updates (Nov 22)**

We have updated Table 4 and the corresponding analysis in Section 4.3 with All Tasks results.

Once again, we would like to thank the reviewers for their constructive feedback that has helped improve the paper significantly!

---

### Author Response · Authors · 2021-03-07
**Camera-Ready Updates**

We thank the AC and reviewers for their insightful feedback. We have updated the camera-ready version. Here is a quick summary of updates:

1. Re-organized the Experiments section to highlight key results related to the paper’s claims.
2. Added additional evaluations in Appendix Sec. F to address R1’s concern regarding the difference between Textworld and Embodied training.
3. Updated the writing to address the points mentioned in R1’s response.
4. Added more related works.
5. The code, data, and models are now publicly available (with pip packages): https://github.com/alfworld/alfworld

Our code release contains: TextDAgger, TextDQN, Seq2Seq agents and more! We look forward to exciting future works in this area.

---

### Decision · Program_Chairs · 2021-01-07
**Final Decision**

**Decision:**

Accept (Poster)

**Comment:**


The paper proposes ALFWorld, which combined TextWorld and ALFRED to create aligned scenarios (one that is text-only, and the other in an embodied visual simulator) so that high-level policies in language can be learned in a simpler world, and then transferred to the embodied one (using the proposed BUTLER architecture).  The proposed BUTLER model consists of three components: 1) a perceptual module (converts environment observation to specification of objects and relations using text), 2) goal-planning module for generating textual specification of subgoals (from observed environment state) and 3) controller module which takes outputs from 1) and 2) and generates a sequence of actions.  Experiments show that using the textual specification, it is possible to models pretrained in the text world can generalize better to embodied settings.

Review Summary: The submission received slightly divergent reviews with R2, R3 recommending acceptance (score 7) and R1 recommending reject (score 4).  All reviewers recognized the novelty of the work, and the potential for follow-up work based on the submission.  After considering the author response and discussion between reviewers, both R2 and R3 agreed that there are indeed flaws with the work as pointed out by R1 (R3 lowered their rating to 6).  Despite the concerns, both R2,R3 remained on the positive side.

Pros:
- The work and proposed framework can stimulate further research on transferring policies from simple text environments to more realistic visual environment. (R2)
- The decomposition of high-level goals into low-level actions sequences is a good direction for future research (R3)
- Good set of experiments and comparisons (R3)
- the paper is clearly written and easy to understand (R2)

Cons:
- The main claim of the work (high-level policies learned in a text-based environment can be transferred to a physically simulated environment) is not properly substantiated by the experimental results. (R1)
- The proposed method is a complex system and simpler baselines should be considered (R2)
- Some assumptions are made in ALFWorld need to be hand designed and may miss important aspects of perception (R3)
- More experiments and ablations are needed to properly evaluate the framework

Despite the issues pointed out by R1, the AC believe that the work can inspire future work in this area, and thus recommend acceptance.  The paper is also well-written and easy to understand.